# Fetal-derived macrophages dominate in adult mammary glands

Norma Jäppinen[1,2], Inês Félix[1,2], Emmi Lokka[1,2], Sofia Tyystjärvi[1,2], Anne Pynttäri[1,2], Tiina Lahtela[1,2], Heidi Gerke[1,2], Kati Elima[1,2], Pia Rantakari[1,2] & Marko Salmi[1,2]

Macrophages serve multiple functions including immune regulation, morphogenesis, tissue homeostasis and healing reactions. The current paradigm holds that mammary gland macrophages first arise postnatally during the prepubertal period from the bone marrow-derived monocytes. Here we delineate the origins of tissue-resident mammary gland macrophages using high-dimension phenotypic analyses, cell-fate mapping experiments, gene-deficient mice lacking selective macrophage subtypes, and antibody-based depletion strategies. We show that tissue-resident macrophages are found in mammary glands already before birth, and that the yolk sac-derived and fetal liver-derived macrophages outnumber the adult-derived macrophages in the mammary gland also in the adulthood. In addition, fetal-derived mammary gland macrophages have a characteristic phenotype, display preferential periductal and perivascular localization, and are highly active in scavenging. These findings identify fetal-derived macrophages as the predominant leukocyte type in the adult mammary gland stroma, and reveal previously unknown complexity of macrophage biology in the breast.

[1] Institute of Biomedicine, University of Turku, Turku, Finland. [2] MediCity Research Laboratory, University of Turku, Turku, Finland. Correspondence and requests for materials should be addressed to M.S. (email: marko.salmi@utu.fi)

Practically all tissues harbor a prominent tissue-resident macrophage population[1–3]. Tissue macrophages are derived from three different developmental origins[2,4–10]. First macrophages arise in the extra-embryonic yolk sac (YS) during early fetal development without monocytic intermediates, and they subsequently colonize most embryonic tissues[11]. Starting from the embryonic day 8.5 (E8.5), macrophage precursors from the YS, and hematopoietic stem cells (HSC) from the hemogenic endothelium, migrate to the fetal liver, and give rise to the first monocytes at E12.5[12,13]. The macrophage descendants of these monocytes then supersede the YS-derived macrophage types in most embryonic tissues. After birth, HSC in the bone marrow produce Ly6C$^+$ monocytes, which also have the potential to migrate to tissues and to differentiate to macrophages[14–16]. The role of the distinct macrophage types in the different macrophage-dependent immunological and non-immunological functions under physiological and pathological conditions remains incompletely understood[1–3].

Mammary gland (MG) derives from ectoderm around E10.5[17]. After formation of buds and sprouting at E13-16, branching leads to the formation of rudimentary ducts by E18. The growth of MG is proportional to the body size until puberty, when estrogen-induced rapid growth and branching takes place. Bone marrow-derived macrophages have been reported to home to MG starting from the age of 2 weeks[18,19]. Diminished numbers of macrophages in osteopetrotic mice (Op/op) mice has been associated with defective collagen I fibrillogenesis, ductal elongation and branching in the MG[18,20,21]. Moreover, macrophages regulate mammary epithelial stem cell activity, epithelial cell proliferation and alveolar budding[22], and hematopoietic cells are needed to support branching morphogenesis in pubertal MG[18]. In breast cancer, the recruitment of macrophages is dependent on CCL2/CCR2 and CSF-1 implying the central role of postnatal monocyte influx in the process[23–28]. Collectively these experiments have led to the prevailing concept that tissue-resident macrophages in MG under physiological and pathological conditions are bone marrow-derived cells of postnatal origin[2,29,30].

Here we evaluate the possible contribution of fetal-derived macrophages to the macrophage pool in the MG. By using several complementary techniques, we show that the majority of tissue-resident macrophages in normal adult MG derive from the YS and fetal liver.

## Results

### Fetal-derived macrophages dominate in the MG at all ages.
Fate-mapping studies have shown that in many tissues F4/80 is a very useful surface marker for discriminating different sub-populations of macrophages[11,14,15]. To study macrophages in MG using multiparameter flow cytometry, we dissociated the tissue and first gated CD45$^+$ cells (leukocytes), then gated CD11b$^+$ cells (macrophages and other myeloid cells) and excluded Siglec-F$^+$ cells (eosinophils), and finally analyzed the expression of F4/80 in this population (Supplementary Fig. 1a, b). We found clearly definable CD45$^+$Siglec-F$^-$CD11b$^+$F4/80$^{Hi}$ (hereafter called F4/80$^{Hi}$) and CD45$^+$ Siglec-F$^-$CD11b$^+$F4/80$^{Int}$ (hereafter called F4/80$^{Int}$) macrophage populations in the inguinal (4$^{th}$) MG of female wild type mice at E16.5 (Fig. 1a). Interestingly, at E16.5, as well as in newborn mice, approximately 50% of MG resident macrophages were F4/80$^{Hi}$ (Fig. 1a). During the fetal development F4/80$^{Hi}$ and F4/80$^{Int}$ phenotypes are indicative of YS-derived and fetal liver-derived macrophages, respectively[2,4,5,11,14,15]. Our analyses at E16.5 thus reveal that MG harbor macrophages before birth, and suggest that both YS and fetal liver contribute to the colonization of macrophages to MG.

During the postnatal development, an F4/80$^{Int}$ and a dominant F4/80$^{Hi}$ macrophage population were clearly identifiable in the prepubertal (1 and 2 wk old), pubertal (4–5 wk old) and adult (3 month old) MG (Fig. 1a). Quantifications of these populations showed increasing numbers of both F4/80$^{Hi}$ and F4/80$^{Int}$ macrophages concomitant with the growth of MG, and a phase of rapid expansion between 2 wk and 5 wk (Supplementary Fig. 2a, b). Both macrophage populations expressed canonical macrophage markers (CD64, MerTK, CD206, C1qa, CSfr1, and Spi) and lacked dendritic cell markers (CD11c, Zbtb46 and Itgax) in flow and qPCR assays (Supplementary Fig. 2c, d). Similar F4/80$^{Hi}$ and F4/80$^{Int}$ macrophage populations were identified in 2$^{nd}$, 3$^{rd}$, and 5$^{th}$ female MG (Supplementary Fig. 2e) and in male MG (Supplementary Fig. 2f). Since F4/80$^{Hi}$ and F4/80$^{Int}$ phenotypes of macrophages in postnatal tissues are indicative of fetal-derived macrophages and adult bone marrow-derived macrophages, respectively[2,4,5,11,14,15], we hypothesize that the fetal-derived macrophages are preserved in the MG throughout the postnatal development at undiminished prevalence.

### YS and fetal liver-derived macrophages persist in MG.
To further analyze the origin of MG macrophages we utilized genetic fate-mapping tools. In the CX3CR1-YFP model[31] tamoxifen administration at E13.5 leads to selective labeling of YS-derived macrophages, since CX3CR1 is not expressed in fetal liver-derived monocytes or their precursors[15]. We found YFP$^+$ cells in the MG of newborn, 2 wk old and 5 wk old CX3CR1-YFP mice, when we administered a single dose of tamoxifen at E13.5 (Fig. 1b). When backgated on the two different CD45$^+$Siglec-F$^-$CD11b$^+$ macrophage populations, we found that the YFP$^+$ cells fell almost exclusively into the F4/80$^{Hi}$ rather than into the F4/80$^{Int}$ macrophage population at all studied time points (Fig. 1b). As a second genetic model we used CSF1R-YFP reporter strain[31], in which tamoxifen administration at E8.5 (before the emergence of the first fetal liver-derived monocytes) leads to the permanent fluorescent conversion of CSF1R$^+$ YS macrophages and their progeny[11]. We observed clear persistence of YS-derived macrophages among F4/80$^{Hi}$ cells in the postnatal MG also in these fate-mapping experiments (Supplementary Fig. 2g).

To selectively ablate YS-derived macrophages (but not fetal liver-derived monocytes/macrophages)[12,32], we injected to wild type mice a single dose of anti-CSF-1R/CD115 antibody at pregnancy day 6.5. The flow cytometric analyses of E17.5 MG verified the complete absence of YS-derived F4/80$^{Hi}$ macrophages, and the normal frequency of fetal liver-derived F4/80$^{Int}$ macrophages after the treatment (Fig. 1c). Depletion of the YS-derived macrophages had no significant effect on the frequency of F4/80$^{Hi}$ macrophage population in MG of 2 and 5 wk old mice (Fig. 1c). Collectively, the cell fate mapping and antibody depletion strategies show that YS-derived macrophages have the F4/80$^{Hi}$ phenotype also in MG, and that they persist in the MG at 5 wk, although they do not constitute a major fetal-derived macrophage population in MG after birth.

To study other fetal-derived macrophage types in MG, we used Plvap$^{-/-}$ mice, which have a selective reduction of fetal liver-derived macrophages in the presence of normal YS- and bone marrow-derived macrophages[33]. We found a significant decrease in the absolute number and frequency of fetal-derived F4/80$^{Hi}$ macrophages in 3 and 5 wk old Plvap$^{-/-}$ mice (Fig. 1d and Supplementary Fig. 2h). The numbers of adult-derived F4/80$^{Int}$ cells in MG remained at the same levels as in littermate wild type controls, although the frequency of F4/80$^{Int}$ cells among all macrophages tended to increase concomitant with the loss of F4/80$^{Hi}$ cells (Fig. 1d and Supplementary Fig. 2h). Collectively these results suggest that fetal liver-derived macrophages

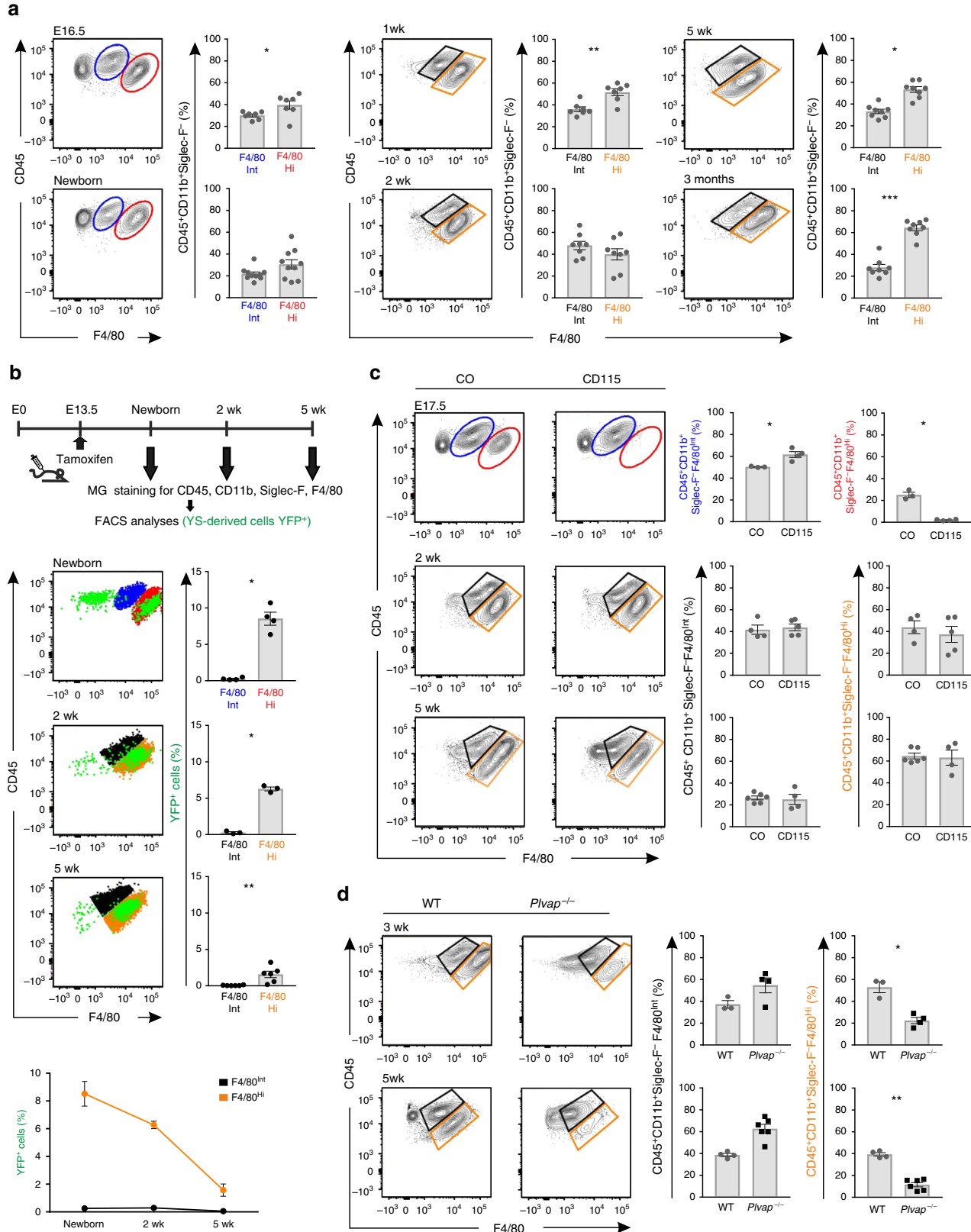

substantially contribute to the pool of tissue-resident macrophages in MG after birth.

**Postnatal monocyte influx to MG is mainly CCR2-independent.** The F4/80$^{Int}$ tissue-resident macrophages found in the postnatal

MG phenotypically resemble bone marrow-derived cells. To genetically analyze the contribution of bone marrow-derived monocytes to the mammary macrophage pool, we used $Ccr2^{-/-}$ and $Nur77^{-/-}$ mice[34,35] manifesting with a major reduction in the numbers of Ly6C$^+$ tissue-infiltrating monocytes, and blood vessel patrolling Ly6C$^-$ monocytes, respectively. We confirmed the

**Fig. 1** Fetal-derived resident macrophages dominate in the mammary gland after birth. **a** Flow cytometric analyses of tissue-resident mammary gland (MG) macrophage populations in wild type (WT) mice. Shown are representative FACS plots and quantifications of F4/80$^{Int}$ and F4/80$^{Hi}$ macrophage populations among CD45$^+$CD11b$^+$Siglec-F$^-$ cells at the indicated prenatal and postnatal time points. **b** Analyses of F4/80$^{Int}$ and F4/80$^{Hi}$ MG macrophages from CX3CR1-YFP reporter mice. The uppermost panel shows the experimental outline (tamoxifen induction at E13.5 and harvesting and staining of MG cells at the indicated time points). The representative FACS plots show backgating of the YFP$^+$ cells (green) onto F4/80$^{Int}$ macrophages (blue in newborn and black in 2 wk and 5 wk old mice) and F4/80$^{Hi}$ macrophages (red in newborn and orange in 2 wk and 5 wk old mice). The quantifications show the frequency of YFP$^+$ cells in each macrophage population. **c** Analyses of MG macrophages in WT mice treated with CD115 or control antibody at E6.5. Shown are representative FACS plots and quantifications of F4/80$^{Int}$ and F4/80$^{Hi}$ macrophage populations at the indicated time points. **d** Analyses of F4/80$^{Int}$ and F4/80$^{Hi}$ MG macrophage populations in Plvap$^{-/-}$ and WT mice at the indicated time points. In all panels (**a**–**d**) MG macrophages were pre-gated as live CD45$^+$CD11b$^+$Siglec-F$^-$ cells. In all panels (**a**–**d**) F4/80$^{Int}$ macrophages have been gated in blue (in embryonic and newborn mice) or in black (in 1 wk – 3 months old mice), and the F4/80$^{Hi}$ macrophages in red (in embryonic and newborn mice) or orange (1 wk – 3 months old mice). In the quantifications, each dot represents one mouse and mean ± SEM are shown. Data are from 3 (**a** E16.5, 1 wk and 5 wk, **d** 5 wk), 2 (**a** newborn, 2 wk and 3 months, **b** newborn and 5 wk, **c** 2 wk, **d** 3 wk) and 1 (**b** 2 wk, **c** E17.5 and 5 wk) independent experiments. *$p < 0.05$, **$p < 0.01$, ***$p < 0.001$ (Kruskal–Wallis test). Source data are provided as a Source Data file

selective reduction of the expected blood monocyte types in adult Ccr2$^{-/-}$ and Nur77$^{-/-}$ mice (Supplementary Fig. 3a; for the gating strategy see Supplementary Fig. 1c).

We found that newborn Ccr2$^{-/-}$ and their wild type control mice had similar frequencies of YS-derived F4/80$^{Hi}$ and fetal liver-derived F4/80$^{Int}$ macrophages in the MG (Fig. 2a). This is consistent with the facts that fetal monocytopoiesis is CCR2-independent[12], and that in newborn Ccr2$^{-/-}$ mice the frequency of Ly6C$^+$ monocytes in the blood was close to that of wild type controls (Supplementary Fig. 3b; for the gating strategy see Supplementary Fig. 1d). In prepubertal (2 wk) and pubertal (5 wk) Ccr2$^{-/-}$ mice the frequency of bone marrow-derived F4/80$^{Int}$ cells in the MG was decreased (Fig. 2a and Supplementary Fig. 3c). Notably, there was only a modest 33 ± 4% reduction in the frequency of F4/80$^{Int}$ population in adult (3 month) Ccr2$^{-/-}$ mice (Fig. 2a). Collectively these data imply that although CCR2-dependent infiltration of bone marrow-derived monocytes does contribute to the pool of tissue-resident macrophages in the MG under homeostatic conditions, Ccr2$^{-/-}$ mice still have a substantial F4/80$^{Int}$ mammary-resident macrophage population at all postnatal time points studied.

To study if Ly6C$^-$ blood-patrolling monocytes, which dominate in migration to primary mammary tumors[31], emigrate to the steady state MG, we used Nur77$^{-/-}$ mice. Although the Ly6C$^-$ monocytes were practically missing from these mice (Supplementary Fig. 3a), the total number and frequency of F4/80$^{Int}$ macrophages were completely normal in the MG of 2 wk and 5 wk old Nur77$^{-/-}$ mice (Supplementary Fig. 3d, e). These findings strongly argue against the contribution of Ly6C$^-$ monocytes to the Ccr2$^{-/-}$ independent infiltration of adult-derived monocytes to the MG. In situ proliferation of fetal-derived macrophages is also an unlikely explanation for the CCR2-independent emergence of F4/80$^{Int}$ cells in adult MG, since bromodeoxyuridine pulse labeling experiments revealed that in 5 wk old wild type mice < 1% of macrophages in the F4/80$^{Hi}$ population undergo cell division (Supplementary Fig. 3f).

To functionally address the recruitment potential of bone marrow-derived macrophages to the MG, we treated 2 wk old mice intravenously with three cycles of alternating anti-CSF-1 antibody and clodronate liposome injections (Fig. 2b). This treatment transiently depletes circulating monocytes and resident macrophages in many tissues[36], and we found that in MG both F4/80$^{Hi}$ and F4/80$^{Int}$ populations were effectively depleted (Fig. 2c). After 11 days, when both Ly6C$^{Hi}$ and Ly6C$^{Low}$ monocyte populations in the blood were recovered (Supplementary Fig. 3g), the frequency of F4/80$^{Int}$ cells in the depleted MG was comparable to those of mock-treated mice, whereas the F4/80$^{Hi}$ macrophages remained virtually undetectable (Fig. 2d). These findings strongly support the notion that F4/80$^{Int}$

macrophages in the postnatal MG represent bone marrow-derived cells. Moreover they indicate that during the recovery period, the bone marrow-derived macrophages are not able to convert to F4/80$^{Hi}$ macrophages in the MG, even when the niche is pre-emptied from fetal-derived macrophages. Collectively our data thus suggest that a significant CCR2-independent influx of bone marrow-derived monocytes takes place in the MG, and that these cells heavily contribute to the F4/80$^{Int}$ tissue-resident macrophage population in the breast after birth.

**High-dimensional mapping of monocytes and macrophages in MG.** Although F4/80 is a well-established phenotypic marker for different macrophage subpopulations, we reasoned that high-dimensional single cell analyses should better discriminate the different myeloid populations. We utilized mass cytometry with a panel of 25 heavy metal-tagged antibodies and the unsupervised t-distributed stochastic neighbor embedding (t-SNE) algorithm to visualize similarities between CD45$^+$ cells on a two-dimensional map. The distinct CD45$^+$ cells were then manually assigned to different leukocyte subpopulations based on the expression of the cell-type selective leukocyte differentiation markers. Using this strategy we visualized several myeloid (Siglec-F$^+$ eosinophils, Ly6G$^+$ granulocytes and Ly6C$^+$/F4/80$^+$/CD64$^+$ monocyte-macrophages) and lymphoid (CD4$^+$ T-helper, CD8$^+$ T-cytotoxic, and B220$^+$ B-cell) populations among the CD45$^+$ cells in the MG of 5 wk and 3 month old wild type mice (Supplementary Fig. 4a-d).

We then performed more detailed analyses of the resident monocytes and macrophages in the CD45$^+$CD11b$^+$ myeloid population (Fig. 3a, b). Two distinct monocyte populations, characterized by high Ly6C expression, were found in the MG of 5 wk old mice (Fig. 3a–c). Both populations were clearly positive for CX3CR1 and F4/80, whereas Siglec-1, CD64 and most notably MHCII were expressed at a lower level on monocyte 1 population than on monocyte 2 population (Fig. 3c). Monocyte population 1 thus resembles recently emigrated monocytes in other tissues[6,37], whereas the phenotype of monocyte population 2 is compatible with monocyte-derived cells already further differentiated towards macrophages.

Among CD45$^+$CD11b$^+$ mammary-resident cells two F4/80$^+$ macrophage populations, clearly distinct from F4/80$^+$Siglec-F$^+$ eosinophils, were identified (Fig. 3a, b). Closer examination of the mass cytometry data revealed that difference in CD206 expression defined these two populations better than F4/80 (Fig. 3a and Supplementary Fig. 5a, b). In 5 wk old mice the CD206$^{Hi}$ macrophage population expressed high levels of CD64, F4/80, and Siglec-1, which are canonical macrophage markers[38], but only low levels of CX3CR1 (Fig. 3d). CD206$^{Neg/low}$ macrophage population expressed lower levels of F4/80, CD64 and Siglec-1 but higher levels of CX3CR1, and in particular of MHCII, than

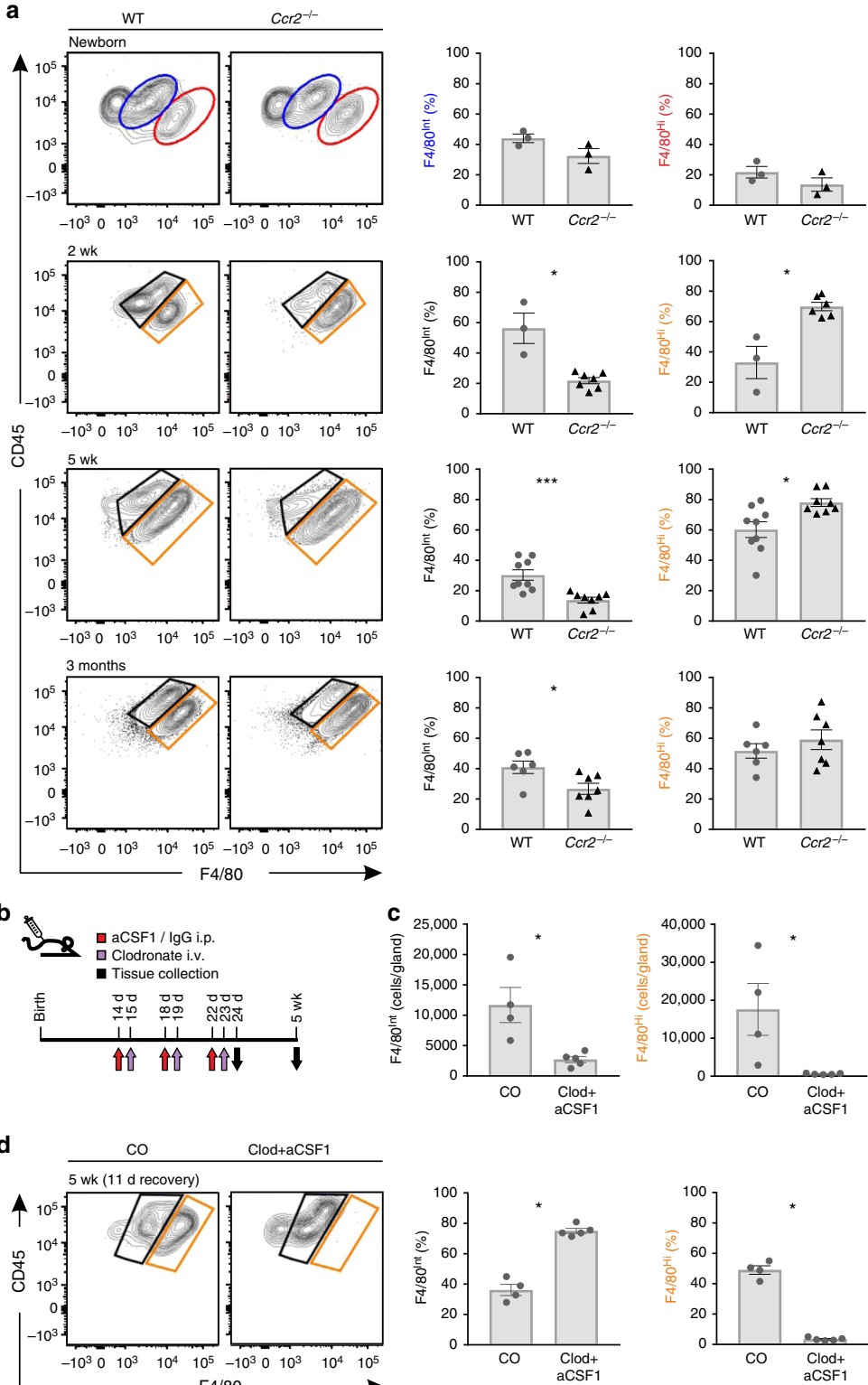

CD206$^{Hi}$ macrophages (Fig. 3d). Fluorimetric analyses revealed MerTK expression in both populations and confirmed the CD64 and MHCII expression level differences between the two populations (Supplementary Fig. 5c). In mass cytometric analyses the phenotypes of the CD206$^{Hi}$ and CD206$^{Neg/low}$ population in 3 month old mice closely resembled those of the 5 wk old mice (Supplementary Fig. 5d). CD206$^{Hi}$ cells represented about 1/3 of all CD11b$^+$ cells and about 2/3 of all macrophages in MG of both 5 wk and 3 month old mice (Fig. 3b). Our mass cytometric data

thus suggest that the fetal-derived tissue-resident macrophages in the MG are defined as F4/80$^{Hi}$CD64$^{Hi}$Siglec-1$^{Hi}$CD206$^{Hi}$ cells, and that CD206 appears to be the most robust single identifier of this subpopulation of macrophages.

We also performed unsupervised clustering analysis of CD45$^+$CD11b$^+$ cells with X-shift, a K-nearest-neighbor density-based clustering algorithm[39]. We subjected leukocyte clusters from MG of 5 wk old mice to a force-directed layout to visualize the spatial relationships between the cell types (defined by the expression of

**Fig. 2** Bone marrow-derived monocytes contribute to the mammary gland macrophage pool after birth. **a** Flow cytometric analyses of mammary gland (MG) macrophage populations in wild type (WT) and $Ccr2^{-/-}$ mice. Shown are representative FACS plots and quantifications of F4/80$^{Int}$ and F4/80$^{Hi}$ macrophage populations at the indicated time points. **b** The experimental outline for depleting macrophages with anti-CSF1 and clodronate treatment. i.p., intraperitoneal; i.v., intravenous. IgG, isotype-matched control antibody. **c** Analysis of total F4/80$^{Int}$ and F4/80$^{Hi}$ macrophage numbers in the MG of clodronate-anti-CSF1 treated (Clod + aCSF1) and control treated (CO) mice one day after the last treatment (at day 24). **d** Analyses of frequencies of F4/80$^{Int}$ and F4/80$^{Hi}$ macrophage populations in MG of Clod + aCSF1 and CO treated mice after an 11 day recovery period (at 5 wk). In **a**, **c**, and **d** MG macrophages were pre-gated as live CD45$^+$CD11b$^+$Siglec-F$^-$ cells. In **a**, **c** and **d** F4/80$^{Int}$ macrophages have been gated in blue (newborn mice) or in black (2 wk – 3 months old mice), and the F4/80$^{Hi}$ macrophages in red (newborn mice) or orange (2 wk – 3 months old mice). In the quantifications, each dot represents one mouse and mean ± SEM are shown. Data are from 1 (**a** newborn), 2 (**a** 2 wk, **c**, **d**) and 3 (**a** 5 wk and 3 months) independent experiments. *$p < 0.05$, **$p < 0.01$, ***$p < 0.001$ (Kruskal–Wallis test). Source data are provided as a Source Data file

characteristic leukocyte subtype markers, Supplementary Fig. 6a) in two-dimensions. We found an apparent hierarchical development suggesting that CD206$^{Neg/low}$ macrophages gradually evolved from the monocyte 1 and 2 populations (Fig. 3e). When similar analyses were done for MG myeloid cells of 3 months old mice, monocyte 2 population was also found to be an intermediate between monocyte 1 population and CD206$^{Neg/low}$ macrophages (Supplementary Fig. 6b, c). These data further supports the notion that CD206$^{Neg/low}$ cells likely represent differentiating tissue-resident cells originating from bone marrow-derived monocytes.

**Functions and localization of CD206$^{Hi}$ macrophages in MG.** In wild type MG, all CD206$^{Hi}$ cells were F4/80$^{Hi}$ in fluorimetric flow cytometry and they expressed high levels of macrophage-selective transcripts *C1qa*, *CSfr1*, and *Spi* (Fig. 4a, b). Treatment of CX3CR1-YFP reporter mice with tamoxifen at E13.5 showed that practically all YS-derived macrophages fell into CD206$^{Hi}$ sub-population (Fig. 4c). Moreover, the CD206$^{Hi}$ subpopulation was heavily underrepresented in the MG of $Plvap^{-/-}$ mice (Fig. 4d) suggesting that fetal liver-derived macrophages are CD206$^{Hi}$. The prenatal origin of CD206$^{Hi}$ cells was also supported by the findings that this population was not decreased in MG of $Ccr2^{-/-}$ mice, and that it was not recovered after clodronate-anti-CSF1 treatments (Fig. 4e, f).

The presence of bone marrow-derived MG macrophages has been associated with ductal branching during puberty[18]. However, we found that the ductal extension and branching at 5 wk were completely intact in $Ccr2^{-/-}$ and $Nur77^{-/-}$ mice, but reduced in $Plvap^{-/-}$ mice (Fig. 5a). These data imply that the presence of fetal-derived macrophages rather than bone marrow-derived macrophages associates with ductal branching morphogenesis.

We finally asked if the fetal-derived and adult-derived MG macrophage subpopulations show differences in scavenging or tissue localization. Flow cytometric analyses showed that CD206$^{Hi}$ cells were superior in scavenging intravenously injected fluorescently-labeled dextran when compared to CD206$^{Neg/low}$ cells in 5 wk old wild type mice (Fig. 5b). We observed that two CD206-independent ligands, acetylated LDL and immune-complexes, were also preferentially taken up by CD206$^{Hi}$ cells (Fig. 5b).

In confocal microscopic analyses we detected both CD206$^+$ and CD206$^-$ cells among F4/80$^+$Siglec-F$^-$ macrophages in MG (Fig. 5c and Supplementary Fig. 7a). In flow analyses only 41 ± 5 % cells in the CD206$^{Neg/low}$ population expressed low levels of CD206, while the rest were completely negative (Fig. 4a). Collectively these data suggest that the vast majority, if not all, CD206$^+$ cells detectable in immunohistochemistry represent CD206$^{Hi}$ macrophages. In MG of 5 wk old wild type mice CD206$^+$ cells were frequently juxtapositioned to the ducts, which were identified by multilayered epithelium surrounded by αSMA$^+$ myofibroblasts (Fig. 5d and Supplementary Fig. 7b). Similarly, CD206$^+$ cells were frequently in contact with CD31$^+$ endothelial

cells of small vessels (Fig. 5e and Supplementary Fig. 7c). Analyses of CD206$^+$ cells in the mice injected intravenously with fluorescent dextran revealed that these macrophages had thin dextran-labeled cytoplasmic projections spanning throughout the whole thickness of vascular wall (Fig. 5f and Supplementary Fig. 7d). Collectively our data indicate that in the MG stroma the CD206$^{Hi}$ macrophages constitute the majority of periductal and perivascular macrophages and that they have intravascular projections enabling efficient scavenging of blood-borne ligands.

## Discussion

Here we show that the majority of tissue-resident macrophages in adult MG are derived from the fetal period. Using cell fate mapping, macrophage depletion strategies, mice genetically deficient for specific macrophage subtypes and high-dimensional single cell expression analyses we show that YS-derived and especially fetal liver-derived macrophages persist in the MG until adulthood. After birth, bone marrow-derived macrophages also infiltrate into the MG under homeostatic conditions using both CCR2-dependent and CCR2-independent mechanisms of migration. Notably, in adult MG the fetal-derived CD206$^+$ macrophages are in direct contact with the ductal tree and blood vessels and they are functionally highly active in scavenging.

We found fetal-derived resident macrophages in the MG already before birth. During puberty and adulthood they were the most prevalent leukocyte type in this organ. The fetal-derived MG macrophages had the F4/80$^{Hi}$ phenotype also characteristic to fetal-derived macrophages in several other organs in adults[2,4,5,11,14,15]. Our high-dimensional mass cytometric analyses complemented these studies by showing that in 3 month old mice the resident F4/80$^{Hi}$ cells were CD206$^{Hi}$MHCII$^{Hi}$CD64$^+$Siglec-1$^+$CX3CR1$^{Low}$. CD206 expression appears to be a useful discriminator for the fetal-derived and adult-derived mammary macrophages, since CD206$^{Hi}$ vs. CD206$^{Neg/low}$ expression divided the populations more robustly than the F4/80$^{Hi}$ vs. F4/80$^{Int}$ expression, and since CD206 is not expressed on tissue-resident eosinophils or other leukocyte types.

The consensus in the field has been that MG macrophages are derived from the bone marrow after birth[2,29,30]. However, to our knowledge, this issue has not been studied in steady state using modern experimental tools. The landmark immunohistochemical analyses in *op/op* mice actually show that in the absence of CSF-1 the number of F4/80$^+$ macrophages around the terminal end buds in 5 wk old mice is reduced by about 50%[18]. Since the development of fetal liver-derived macrophages, as opposed to YS-derived and bone marrow-derived macrophages, are not dependent on the CSF-1R signaling[12,32], we propose that the residual macrophages observed in *op/op* MG are fetal liver-derived cells. In breast cancer new monocyte-derived macrophages infiltrate to the MG as the cancer progresses[23,24]. However, cancer represents an inflammatory condition, and it is well accepted that bone marrow-derived inflammatory monocytes readily infiltrate tissues under such conditions. In fact, when we experimentally generated an empty

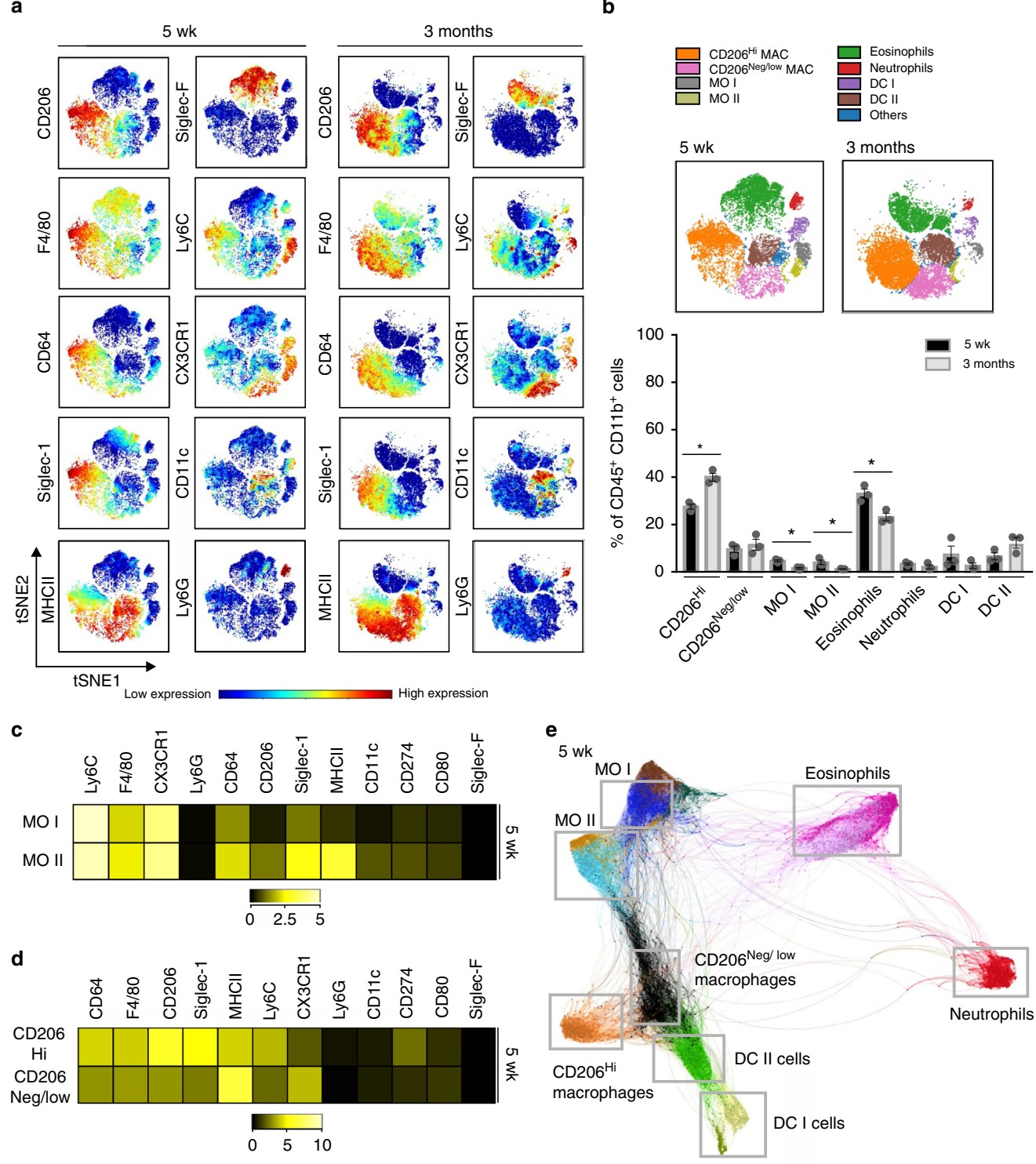

mammary niche resembling the conditions of de novo emerging tumors[25–28] or irradiated mice[18], we observed a vigorous infiltration by bone marrow-derived monocytes. Interestingly, unlike in most other tissues[34,37,40,41], the bone marrow-derived monocytes used largely CCR2-independent mode of migration to infiltrate the MG. Since monocyte-derived macrophages are plastic cells[6,10], it remains possible that in an experimentally emptied MG niche, and perhaps even in a normal MG niche, the adult bone marrow-derived monocytes could gradually start to differentiate to cells phenotypically and functionally resembling the depleted

fetal-derived macrophages. Therefore, our results do not contradict the importance of tissue environment in shaping macrophage identity[7], but our data now for the first time show that the fetal-derived macrophages dominate in the MG at postnatal time points under strictly homeostatic conditions.

In conclusion, our data show that the resident MG macrophages originate from three ontogenically different sources, and that under steady state the fetal-derived macrophages dominate even in the adult MG. Our findings have important implications for re-evaluating the contribution of different macrophage

**Fig. 3** Mass cytometry identifies the diversity of myeloid leukocytes in the mammary gland. **a** Unsupervised analyses of randomly sampled $CD45^+CD11b^+$ myeloid cells from mammary glands (MG) of 5 wk and 3 months old wild type (WT) mice. The expression intensity of the indicated markers is displayed on the t-SNE maps. **b** Frequencies of resident $CD45^+CD11b^+$ myeloid cells in the MG in steady state. Based on the unsupervised expression analyses of all leukocyte subtype-selective markers (**a**), coloring was manually added to the t-SNE maps to visualize different cell populations. The frequencies of each cell type from all $CD45^+CD11b^+$ leukocytes is shown as mean ± SEM (each dot represents one mouse). *$p < 0.05$, **$p < 0.01$, ***$p < 0.001$ (Kruskal–Wallis test). MAC macrophages, MO monocytes, DC dendritic cells. **c** Mass cytometric analysis of mean marker expression for the two resident $Ly6C^+$ monocyte (MO) subsets in the MG of 5 wk old WT mice. **d** Mass cytometric analysis of mean marker expression for $CD206^{Hi}$ and $CD206^{Neg/low}$ macrophage subsets in the MG of 5 wk old WT mice. **e** Single-cell force-directed layout of MG $CD45^+CD11b^+$myeloid cells of 5 wk old WT mice. The cell clusters were generated using unsupervised hierarchial clustering. Based on the expression analyses of all leukocyte subtype-selective markers, gray boxes and coloring were manually added to visualize the identity of different clusters. All data are from 3 independent experiments. Source data are provided as a Source Data file

subpopulations to the physiology and immunology of developing MG and for the development of monocyte/macrophage targeting therapies in breast cancer.

## Methods

**Mice**. Plvaptm1Salm (*Plvap*$^{-/-}$) mice have been previously described[33,42]. They have completely normal production of YS- and bone marrow-derived macrophages, but show a selective reduction in the number of fetal liver-derived macrophages in different tissues due to a defective exit of fetal liver monocytes to the blood. *Ccr2*$^{-/-}$ (stock 004999), *Nur77*$^{-/-}$ (stock 006187), R26R-EYFP (stock 006148), Cx3cr1-CreERT2 (stock 020940) and Csf1r-Mer-iCre-Mer (stock 019098) mice were purchased from Jackson Laboratories. The exit of bone marrow-derived $Ly6C^{Hi}$ monocytes to blood is dependent on CCR2 signaling, and consequently *Ccr2*$^{-/-}$ mice have a major reduction in the number of "inflammatory" tissue-infiltrating monocytes[34,40,43]. The conversion of $Ly6C^{Hi}$ monocytes to blood vessel patrolling $Ly6C^{Low}$ monocytes is partially dependent on the transcription factor Nur77, and *Nur77*$^{-/-}$ mice manifest with a severe reduction in the $Ly6C^{Low}$ monocytes in the blood[35]. C57BL/6J and C57BL/6N mice were purchased from Janvier labs.

Mice were housed under specific pathogen free conditions and a 12 h light-dark cycle at a temperature of 22 °C in the animal facilities of University of Turku (Turku, Finland). The mice had free access to chow (irradiated standard pellets) and reverse osmosis water. All animal experiments were formally reviewed and approved by the Ethical Committee for Animal Experimentation in Finland. Experiments were carried out in adherence with the rules and regulations of the Finnish Act on Animal Experimentation (497/2013) and in accordance to the 3R-principle under Animal license numbers 5587/04.10.07/2014 and 6211/04.10.07/2017. Mice were used at the indicated ages. Sex-matched wild type mice were used as controls in each experiment.

**Timed pregnancies and tamoxifen administration in utero**. For studying YS-derived macrophages Csf1r-Mer-iCre-Mer mice were crossed with ubiquitously fluorescent R26-EYFP mice. Embryonic development was estimated considering the day of vaginal plug appearance as embryonic age of 0.5 days (E0.5). For induction of reporter recombination in the offspring, a single dose of tamoxifen (1.5 mg, mented with 0.75 mg progesterone) was injected i.p. to the pregnant females at E8.5, before the emergence of the first fetal liver-derived monocytes. This leads to the permanent fluorescent conversion of CSF1R+ YS macrophages and their progeny[11]. As a positive control, we analyzed the converted cells in the spleen and found them solely in the $CD45^+CD11b^+F4/80^{Hi}$ population known to represent YS-derived macrophages[11,12].

In other experiments Cx3cr1-CreERT2 male mice were crossed with R26-EYFP female mice, and the pregnant females were treated with tamoxifen and progesterone at E13.5 to induce reporter recombination. This induction protocol also leads to selective labeling of YS-derived macrophages, since CX3CR1 is not expressed in fetal liver-derived monocytes or their precursors[14,15].

**Macrophage depletion**. To deplete YS-derived macrophages pregnant female mice were treated with a single injection of CSF-1R blocking antibody (clone AFS98, Bio X Cell) or rat IgG2a control antibody (clone 2A3, Bio X Cell). Three µg of the antibodies in sterile PBS were administered i.p. to pregnant females at E6.5, as described[12,32]. Mice were sacrificed at E17.5 or at postnatal age of 2 wk or 5 wk for flow cytometric analyses.

To deplete tissue-resident macrophages after birth, 2 wk old C57Bl/6N mice were cyclically treated with anti-CSF1 antibody and clodronate (Fig. 2b). To that end, three doses of CSF1 neutralizing antibody (Clone 5A1, BioXcell) or control IgG (clone HRPN, BioXcell) were given i.p. (0.5 mg on postnatal day 14, 0.25 mg on postnatal day 18 and 0.25 mg on postnatal day 22). On subsequent days, three doses of clodronate or control liposomes (Liposoma) 50 µl/injection on postnatal days 15, 19 and 23) were administered i.v. The mice were sacrificed 1 or 11 days after the final clodronate treatment. In control experiments using kidney, we saw a full recovery of a known bone marrow-derived $CD11b^+F4/80^{Int}$ macrophage

population, but not that of a known fetal-derived $CD11b^{Int}F4/80^{Hi}$ macrophage population, verifying the robustness of the model.

**Flow cytometry and cell sorting**. Unless otherwise stated, the 4$^{th}$ MG from virgin females was collected, inguinal lymph node was removed and the fat pad was finely minced with scissors in Hank's Buffered Saline (Sigma-Aldrich, H9394). MG was digested with 1 mg/ml collagenase D (Roche, 1108886601) and 50 µg/ml DNase 1 (Roche, 10104159001) for 60 min at +37 °C. Cell suspension was filtered through silk (pore size 77 µm). The cell suspensions were incubated with purified anti-CD16/32 (clone 2.4G2; Bio X Cell) for 10 min on ice to block Fc-receptors. Immunofluorescence stainings were performed at 4 °C for 20 min (the antibodies used are listed Supplementary Table 1). Flow cytometry was performed with a LSR Fortessa flow cytometer (Becton Dickinson), and data were analyzed using the FlowJo software (FlowJo LLC).

For proliferation analysis, mice were injected i.p. with 120 µl of 10 mg/ml solution of bromodeoxyuridine (BrdU, BD Bioscience) 2 h before the sacrifice. After immunofluorescence stainings cells were fixed, and stained with FITC–conjugated BrdU antibody (BrdU Flow Kit, BD Bioscience).

For qPCR analysis, $CD45^+Siglec-F^-CD11b^+F4/80^{Int}$ and $CD45^+Siglec-F^-CD11b^+F4/80^{Int}$ or $CD45^+Siglec-F^-CD11b^+F4/80^+CD206^{Neg/low}$ and $CD45^+Siglec-F^-CD11b^+F4/80^+CD206^{Hi}$ cells from MG of 5 wk old mice were sorted using a FACS aria II cell sorter (100 µm nozzle, Beckton-Dickinson). The purity of the isolated populations was >95%.

**Gating strategies for flow cytometry**. The gating strategies for fetal and adult MG macrophages and fetal and adult blood monocytes are depicted in Supplementary Fig. 1. All cells were pre-gated as viable. For the fetal and adult MG macrophages, leukocytes were identified as $CD45^+$ cells, and then Siglec-F$^+$ cells (i.e. eosinophils) were excluded, and $CD11b^+$ myeloid cells selected for analyses of F4/80 (Figs 1, 2) and CD206 (Figs. 4, 5). Based on the previous literature[2,4,5,11,14,15] and our current results, F4/80 expression level was used as a phenotypic surrogate of the ontogenic origin of the macrophage populations. Thus, during the fetal development, YS -derived macrophages are F4/80$^{Hi}$, whereas fetal liver-derived macrophages are F4/80$^{Int}$. After birth, both fetal-derived macrophage subtypes become F4/80$^{Hi}$, while F4/80$^{Int}$ cells represent bone marrow-derived macrophages. For visualization, F4/80 was plotted against CD45 in the initial analyses (Figs. 1, 2) to allow maximal separation of F4/80$^{Hi}$ and F4/80$^{Int}$ populations. The frequency of the different macrophage subpopulations was determined from $CD45^+Siglec-F^-CD11b^+$ leukocytes.

In selected experiments additional antibodies (CD64, MerTK, MHCII or CD11c) were added to the pool of CD45, Siglec-F, CD11b, and F4/80 antibodies, or to the pool of CD45, Siglec-F, CD11b, F4/80, and CD206 antibodies for further phenotyping of the different macrophage populations (Supplementary Fig. 1a). Similarly, in the proliferation experiments, the incorporated BrdU was detected with an anti-BrdU antibody added to the macrophage phenotyping pool of antibodies.

In the fate-mapping experiments, MG cells were also stained with the macrophage phenotyping pool of antibodies leaving the 488-channel empty. The YFP signal from the genetically converted YFP$^+$ cells was simultaneously recorded from the 488-channel, and then backgated onto the populations defined with the macrophage phenotyping antibodies.

Fetal blood monocytes were defined as $CD45^+CD11b^+$ and adult blood monocytes as $CD45^+CD115^+CD11b^+$ cells for analyzing Ly6C expression (Supplementary Fig. 1).

When indicated, the total number of a given macrophage subtype per MG was determined by dividing the number of events analyzed in flow cytometry by the volume fraction of the given MG sample subjected to the flow analyses.

**Mass cytometry**. The cells were isolated as described above for flow cytometry. They were first stained with 2.5 µM Cell-ID Cisplatin (Fluidigm; cat. 201064) at room temperature for 5 min to exclude dead cells. After washings and Fc-blocking with anti-CD16/32 the cells were stained with a heavy-metal isotope–labelled mAb cocktail (Supplementary Table 1) for 30 min at room temperature. After washings,

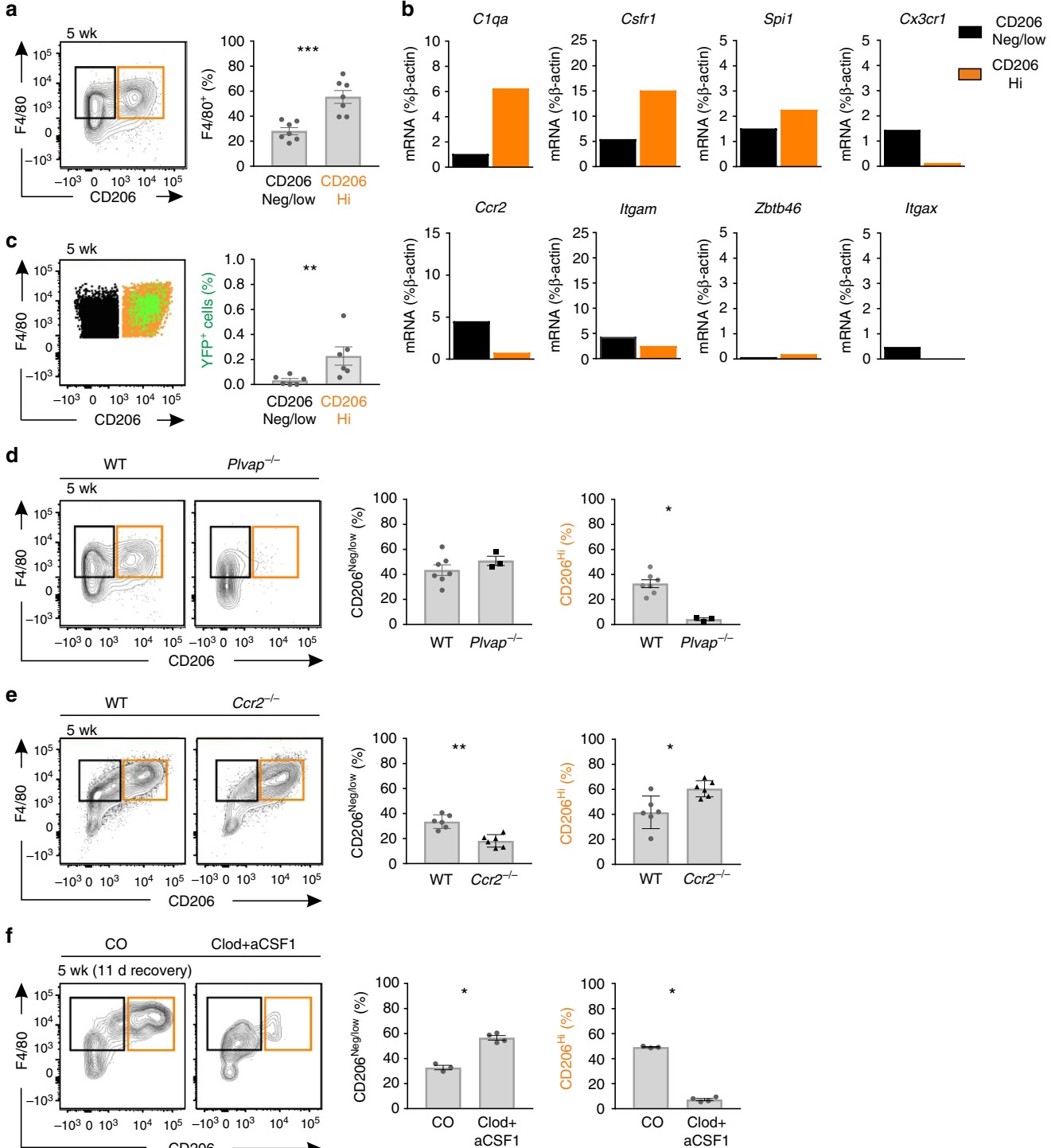

**Fig. 4** CD206[Hi] leukocytes represent fetal-derived macrophages in adult mammary glands. **a** Fluorimetric flow analyses of 5 wk old wild type (WT) mammary gland (MG) macrophages for F4/80 and CD206. **b** qPCR analysis of the indicated transcripts in CD206[Neg/low] and CD206[Hi] MG macrophage populations sorted from 5 wk old WT mice. The sorted populations were pooled from five donors, and the mRNA expression values are shown in relation to a control gene β-actin. **c** Analyses of CD206[Neg/low] and CD206[Hi] MG macrophages from 5 wk old CX3CR1-YFP reporter mice (tamoxifen induction at E13.5). The representative FACS plot shows backgating of the YFP[+] cells (green) onto CD206[Neg/low] (black) and CD206[Hi] (orange) cells, and the quantification shows the frequency of YFP[+] cells in these two macrophage populations. **d** Fluorimetric analyses of CD206[Neg/low] and CD206[Hi] MG macrophages in 5 wk old WT and *Plvap*[−/−] mice. **e** CD206[Neg/low] and CD206[Hi] MG macrophages in 5 wk old WT and *Ccr2*[−/−] mice. **f** CD206[Neg/low] and CD206[Hi] MG macrophages in control (CO) and clodronate-anti-CSF1 (Clod + aCSF1) treated mice at 5 wk (after an 11 day recovery period). In all panels (**a**–**f**) MG macrophages were defined as CD45[+]CD11b[+]Siglec-F[−]F4/80[+] cells, and CD206[Neg/low] cells have been gated in black and CD206[Hi] cells in orange. In the quantifications, each dot represents one mouse and mean ± SEM are shown. Data are from 3 (**a**), and 2 (**b** (for sortings), **c**–**f**) independent experiments. *$p < 0.05$, **$p < 0.01$, ***$p < 0.001$ (Kruskal–Wallis test). Source data are provided as a Source Data file

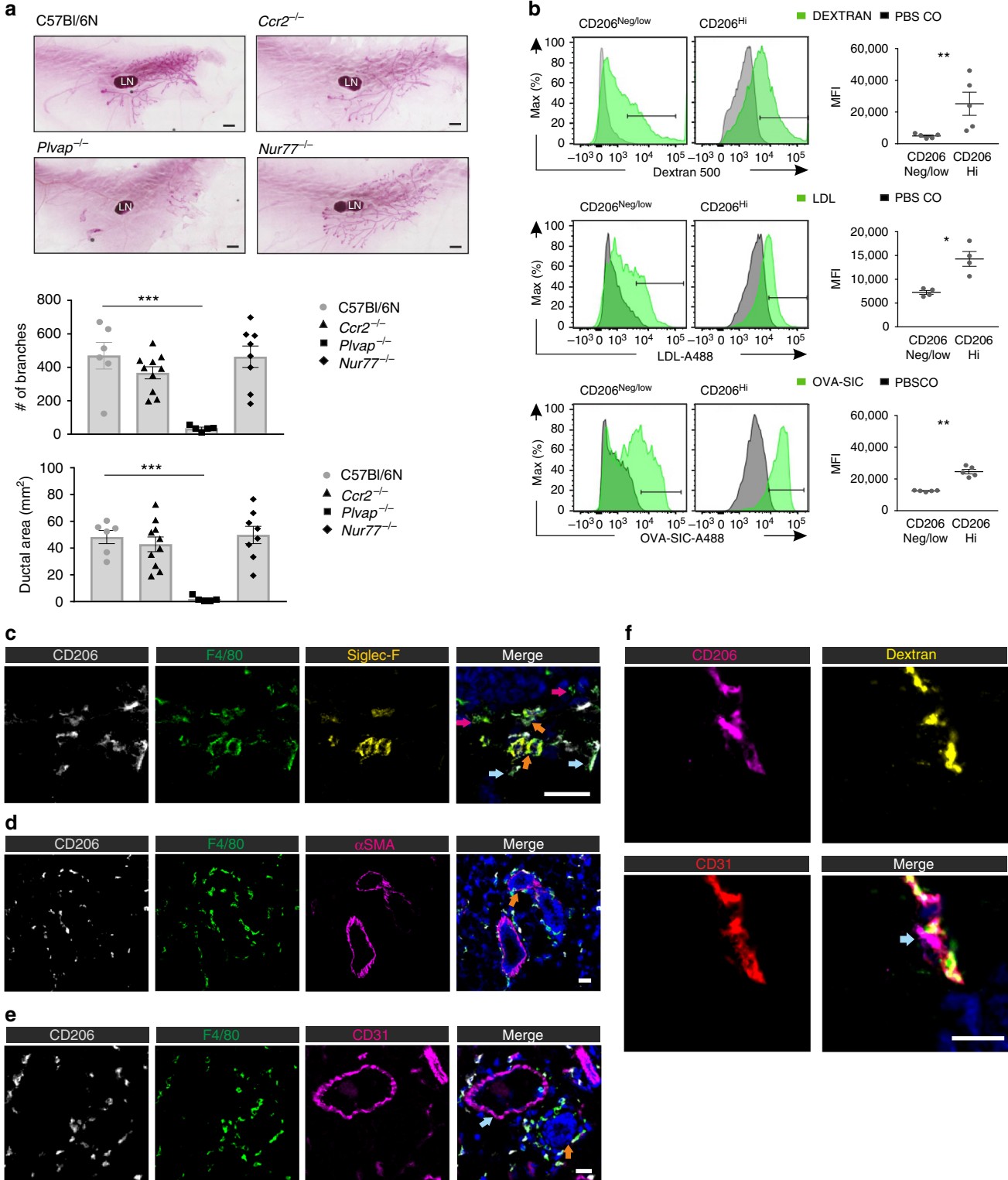

the cells were incubated for 1 h with DNA intercalation solution (1:1000 Cell ID Intercalator-103Rh in MaxPar® Fix and Perm Buffer; cat. 201067; Fluidigm) at room temperature, fixed with 4% paraformaldehyde solution (PFA; Santa Cruz Biotechnology; cat. sc-281692) overnight and then pelleted. The cells were resuspended to MaxPar Water (cat. 201069; Fluidigm) immediately prior to data acquisition with a CyTOF mass cytometer (Helios, Fluidigm).

Bead normalized mass cytometry data were gated for live (Cisplatin⁻) singlets ($^{191}Ir^+$), and applied the t-SNE algorithm to analyze and visualize the multidimensional data in two-dimensions (viSNE,Cytobank, https://www.cytobank.org). A proportional event sampling using 18,000–25,000 events per sample in CD45 analysis, and 3000–14,000 events in CD11b analysis, was selected.

The distinct CD45⁺ cell populations were then manually assigned to different leukocyte subpopulations based on the expression of the cell-type selective leukocyte differentiation markers.

In certain experiments (Supplementary Fig. 5a) the mass cytometry data was manually analyzed using bi-axial blottings to allow direct comparison with the fluorimetric data.

Alternatively, the data were uploaded to the VorteX clustering environment (https://github.com/nolanlab/vortex/releases/tag/29-Jun-2017). 10000 events were imported and all files were merged into one dataset (different datasets by the age). Fifteen parameters were selected for unsupervised hierarchial clustering with X-shift algorithm. For the clustering the default settings were used with nearest

**Fig. 5** CD206[Hi] macrophages show distinct functions and location in adult mammary glands. **a** Whole mount carmine alumine stainings of mammary gland (MG) and quantification (branch number and ductal area) of the branching morphogenesis in 5 wk old mice of the indicated genotypes. LN, lymph node. **b** Uptake of intravenously administered fluorescent 500 kDa dextran, acetylated LDL and ovalbumin-anti-ovalbumin antibody immunocomplexes by CD206[Neg/low] and CD206[Hi] macrophage subpopulations in the MG of 5 wk old wild type (WT) mice. Control mice received PBS injections. MFI, specific mean fluorescence intensity. **c** Immunohistochemical analysis of CD206, F4/80, and Siglec-F expression in the MG of 5 wk old WT mice. Representative CD206[+]F4/80[+]Siglec-F[−] macrophages (light blue arrows), CD206[−]F4/80[+]Siglec-F[−] macrophages (magenta arrows) and CD206[−]F4/80[+]Siglec-F[+] eosinophils (orange arrows) are pointed out. Blue is Hoechst. Bar, 50 μm. **d** Analysis of CD206, F4/80 and alpha smooth muscle actin (αSMA) expression in the MG. An orange arrow points to a representative duct. Blue is DAPI. Bar, 25 μm. **e** Analysis of CD206, F4/80, and CD31 expression in the MG. A light blue arrow points to a representative vessel, and an orange arrow to a representative duct. Blue is DAPI. Bar, 25 μm. **f** Microscopic analyses of CD206 and CD31 in MG of mice injected intravenously with fluorescent dextran (yellow). Transmural projections of CD206[+] macrophages, which have ingested dextran, are pointed out by a light blue arrow. Bar, 10 μm. In **b** MG macrophages were defined as CD45[+]CD11b[+]Siglec-F[−] cells. In the quantifications, each dot represents one mouse and mean ± SEM are shown. Quantitative data are from three (**a**, **b** OVA-SIC) and two (**b** Dextran and LDL) independent experiments. **c–f** are representative figures from three independent experiments *p < 0.05, **p < 0.01, ***p < 0.001 (Kruskal–Wallis test). Source data are provided as a Source Data file

density estimation (K) from 150 to 10, with 30 steps. Thereafter all data sets were selected and the elbow point (K) was calculated (K = 61 for 5 wk old and K = 94 for 3 months mice). All 13 clusters were selected and the force-directed layout was created (ForceAtlas2 algorithm; all cell events from clusters smaller than 1000 events or 1000 randomly selected events from the clusters bigger than 1000 events). The distinct CD45[+] cell clusters were then manually assigned to different leukocyte subpopulations based on the expression of the cell-type selective leukocyte differentiation markers. The layout and the visualization were produced with Gephi 0.9.1 (https://gephi.org). Finally, two-dimensional figures were produced from the original three-dimensional data.

**qPCR**. Total RNA was isolated from sorted mammary macrophages of 5 wk old wild type female mice using the RNeasy Plus Micro kit (QIAGEN). The RNA was reverse transcribed to cDNA with SensiFast™ cDNA Synthesis Kit (Bioline) according to the manufacturer's instructions. Quantitative PCR was carried out using Taqman Gene Expression Assays (ThermoFisher Scientific) for *Cx3cr1* (Mm00438354_m1), *Itgam* (Mm00434455_m1), *Ccr2* (Mm04207877_m), *Csf1r* (Mm01266652_m1), *Zbtb46* (Mm00511327_m1), *C1qa* (Mm00432142_m1), *Spi1* (Mm00488142_m1), *Itgax* (Mm00498701_m1) and *Actb* (Mm02619580_g1; control gene). The reactions were run using either the QuantStudio12K Flex Real-Time PCR system (Thermo Fisher Scientific) or Quant Studio 3 Real-Time PCR System (Thermo Fisher Scientific). Relative expression levels were calculated using Applied Biosytems® analysis modules in Thermo Fisher Cloud computing platform (ThermoFisher Scientific). The results were presented as percentages of the control gene mRNA levels from the same samples.

**Analysis of ductal branching in MG**. The 4th MG was mounted onto a glass slide, left to adhere briefly, and fixed in Carnoy's fixative (60% ethanol, 30% chloroform, 10% glacial acetic acid) overnight at +4 °C or for 4 h at room temperature. The MG was rehydrated in decreasing ethanol series, and stained with carmine alum (STEMCELL technologies, cat 07070) according to the manufacturer's instructions. The samples were dehydrated, cleared in xylene for 2–3 days, and mounted with DPX Mountant (Sigma-Aldrich). Imaging was performed with Axiovert M200 microscope using a 5 × /0.25 objective. The MG area covered by the ductal tree and the number of ductal branches were tracked manually, and quantified using ImageJ with 'Skeletonize2D/3D' and 'AnalyzeSkeleton' plugins.

**In vivo uptake experiments**. Immune complexes were prepared in vitro by incubating ovalabumin (OVA)- Atto488 (41235 Sigma, 2 mg/ml in PBS) at 5:1 molar ratio with rabbit polyclonal anti-OVA IgG for 1 h at 4 °C[44]. 100 μl of OVA-SIC, or PBS as a vehicle control, was administrated via tail vein injections to 5 wk old wild type mice. In other experiments, 10 μg of fluorescently labeled acetylated low-density lipoprotein (LDL; Alexa Fluor 488 conjugated, L23380 Thermo Fisher Scientific) or 0.8 mg of 500 kDa Dextran (fluorescein, D7136 L23380 Thermo Fisher Scientific) were administered i.v. in a 150 μl volume. The recipient mice were sacrificed after 1 h (dextran) or 2 h (LDL and OVA-SIC), and the 4th MG were collected for flow cytometry. The MFI for each label in the PBS-treated controls was subtracted from MFI for the corresponding label in the test compound-treated mice to obtain the specific MFI for each test compound.

**Immunofluorescence stainings and confocal imaging**. The 4th MG was collected, embedded in optimal cutting temperature compound (OCT; Tissue-Tek cat.4583), and snap-frozen with liquid nitrogen. 6–8-μm sections were cut using a cryomicrotome (Leica). The sections were fixed with ice-cold acetone (Merck, Cat. 1.00014.2500), blocked with 5% normal goat serum (Jackson ImmunoResearch, Cat. 055-000-121), and stained with the antibodies listed in the Supplementary Table 1 for 30 min at room temperature. After washing with PBS (Gibco, cat. 18912-014) the sections were mounted with ProLong Gold Antifade Mountant with DAPI (4',6-diamidino-2-phenylindole; Thermo Fisher Scientific, cat. 62249),

or the nuclei were stained with Hoechst (Thermo Fisher Scientific, Cat. 62249), and then mounted with ProLong Gold Antifade Mountant without DAPI (Thermo Fisher Scientific, cat. P36930). Images were acquired using a spinning disk confocal microscope (Intelligent Imaging Innovations) with a Plan-Apochromat 20 × /0.8 or 10 × /0.45 objective or LSM780 confocal microscope (Zeiss) with C-Apochromat 40 × /1.2 or C-Apochromat 63 × /1.2 oil objective. Background subtractions, linear brightness and contrast adjustments, and median filtering for reduction of noise were performed with ImageJ software.

**Statistical analyses**. Adult mice were allocated to experimental groups without specific randomization methods, because comparisons involved mice of distinct genotypes. The investigators were blinded to the genotype of the embryos during the experimental procedures. Numeric data are given as mean ± SEM. Comparisons between the genotypes or treatment groups were done using Kruskal–Wallis test. SAS 9.4 (SAS institute, Inc.) was used for the statistical analyses. P-values < 0.05 are considered to be statistically significant.

**Reporting summary**. Further information on experimental design is available in the Nature Research Reporting Summary linked to this article.

## Data availability
The data that support the findings of this study are available from the corresponding author upon reasonable request. A reporting summary for this Article is available as a Supplementary Information file. The source data underlying Figs. 1a–d, 2–d, 3b, 4a–f, 5a, b and Supplementary Figures 2a-h, 3a-g, 4c and 5b are provided as a Source Data file.

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

## Acknowledgements

We thank Ms. Etta-Liisa Väänänen for expert technical help and M.Sc. Kristiina Santalahti for help with the statistics. This study was financially supported by the Academy of Finland, Sigrid Juselius Foundation, Aatos and Jane Erkko Foundation and Instrumentarium Science Foundation.

## Author contributions

N.J. performed and analyzed flow cytometric studies and whole mount stainings, I.F. participated in mass cytometry, E.L. did scavenging assays, S.T. performed microscopy and K.E. analyzed qPCR data. A.P., T.L., and H.G. assisted in animal studies. P.R. planned, performed, and supervised experiments and analyzed data. M.S. conceptualized the study, analyzed data, and wrote the manuscript.
