## [Peer Review File · Nature Communications]

Reviewers' comments:

Reviewer #1 (Innate immunity, macrophage biology, inflammation)(Remarks to the Author):

"Fetal liver macrophages dominate in adult glands"
Nature communication review comment,

General comments,

"Fetal liver macrophages dominate in adult glands" by Professor Marko Salmi and co-authors, this study implied that the resident macrophages in mammary glands were derived from three different sources. Also they stated that macrophages from fetal liver dominate in the adult mammary gland. This concept may be interesting. Although the research field of differentiation system in macrophage is an important area in immunological field, the authors need to more carefully examine characterization of macrophages in mammary gland. The following comment should be addressed.

comments;

The authors showed FACS data in this manuscript. To distinguish macrophages from different sources, two markers such as CD45 and F4/80 were used. Indeed, F4/80 is a famous and well-characterized marker for definition of macrophage. In contrast, CD45 is a marker for all of haematopoietic cells. In this manuscript, fractionation of macrophages was very important to prove the authors' argument. However, the classification of the cell was not done accurately. Such prejudiced analysis of FACS data mislead the reader. To classify resident macrophages in mammary gland, in addition to F4/80, suitable marker(s) such as Mac1, CD206 or CD115 after gating by CD45 was used here for macrophage definition. In all of FACS data, such analysis should be performed and the authors should consider the conclusion again. This is simple experiment.

Also, gating strategy of FACS analysis is also incorrect. (this is caused by using fewer markers). For example, in fig1d, authors stated that they showed F4/80 int and F4/80 hi macrophage population in mammary gland in WT and Plvap^{-/-} mice. However, in the fraction of F4/80 int (black line), F4/80hi population was included. Similar ambiguous analysis was performed in Fig. 2d. Since theme of this manuscript is differentiation system of macrophages in mammary gland, a precise FACS analysis should be performed.

Reviewer #2 (CyToF, Systems immunology, immune cell subsets)(Remarks to the Author):

In this study, the authors use a large variety of mouse model systems to study the origins of mammary gland resident monocyte/macrophage populations. They show that F4/80hi cells are enriched for fetal derived macrophage cells that can be better defined using CD206 as marker based on mass cytometry data analysis performed. This appears to be one of the most rigorous studies of

mammary gland resident monocyte macrophage steady-state ontogeny and should be broadly useful. Overall, my impression is that this is a well-executed study and I do not have any major problems.

Comments:

On line 65, the justification on the origins of these cell populations are not provided at this point. Instead perhaps objective descriptions should be used and origins stated as a hypothesis?

Paragraph starting on line 83. If possible a reference for the validity of this fate mapping model should be provided.

In general, figure legends need more details to allow them to be properly interpreted. Fig. 1. All upstream gating should be described in figure legends for each panel. Panel b, the colors and backgating strategies are not at all described. What is the basis for coloring in Fig. 3e? Are these manual or automatically defined clusters?

Error in panel label on Fig S3b – one panel label missing in the pdf I have.

When mass cytometry is introduced, it would be easier to follow if similar gating strategies on the previously quantified F4/80^{int} and hi cells were annotated on the tSNE plots in addition to these additional subsets defined in Fig. 3b. To make this part of the paper interpretable in light of the data shown in Fig. 1 and 2, it is important that equivalent gating strategies are used for mass cytometry as for the quantifications of the fluorescent data shown in previous figures. Although this does make sense and subsequent figures support the utility of CD206 as a marker for embryonic-derived macrophages, biaxial dot plots (with mass cytometry F4/80 staining) should then be used to support lines 173-174.

Although it is nice that the mass cytometry data could be used as a justification for the use of CD206, I wonder if this could have been deduced in a more objective way from these data. For instance, perhaps the authors could use high dimensional clustering analysis (as in Fig. 3e) to first define the most-differentiated macrophage, then use an automatic gating algorithm to objectively determine the most accurate way to define these cells. This is just a suggestion and not necessary for the validity of this manuscript.

For Fig. 3B, biaxial scatter plots should be shown for the gating strategies used for each of these manually defined subsets.

For the recovery of F4/80^{hi} cells after anti-CSF-1 + clodronate liposome treatment (figs 2,4), did the authors go beyond 11 days of recovery to see if these cells can eventually recover?

Reviewers' comments:

Reviewer #1 (Innate immunity, macrophage biology, inflammation)(Remarks to the Author):

We would like to thank the reviewer for the positive and constructive comments.

“Fetal liver macrophages dominate in adult glands”
Nature communication review comment,

General comments,

“Fetal liver macrophages dominate in adult glands” by Professor Marko Salmi and co-authors, this study implied that the resident macrophages in mammary glands were derived from three different sources. Also they stated that macrophages from fetal liver dominate in the adult mammary gland. This concept may be interesting. Although the research field of differentiation system in macrophage is an important area in immunological field, the authors need to more carefully examine characterization of macrophages in mammary gland. The following comment should be addressed.

comments;

The authors showed FACS data in this manuscript. To distinguish macrophages from different sources, two markers such as CD45 and F4/80 were used. Indeed, F4/80 is a famous and well-characterized marker for definition of macrophage. In contrast, CD45 is a marker for all of haematopoietic cells. In this manuscript, fractionation of macrophages was very important to prove the authors' argument. However, the classification of the cell was not done accurately. Such prejudiced analysis of FACS data mislead the reader. To classify resident macrophages in mammary gland, in addition to F4/80, suitable marker(s) such as Mac1, CD206 or CD115 after gating by CD45 was used here for macrophage definition. In all of FACS data, such analysis should be performed and the authors should consider the conclusion again. This is simple experiment.

We sincerely apologize for describing our gating strategy too superficially in the Methods and Figure legends of the original submission. In all mammary gland FACS analyses we have first gated leukocytes as CD45⁺ cells, and then Siglec-F⁺ cells (i.e. eosinophils) have been excluded. From the CD45⁺Siglec-F⁻ cells both CD11b and F4/80 (Figs. 1 and 2), or CD11b and CD206 (Figs.4 and 5) have been used as markers of macrophages. Thus in all analyses, the definition of macrophages relies on the use of multiple markers, just as the reviewer correctly recommends.

We have now specified the macrophage gating more precisely both in the main text and in the Figure legends. In Fig. 1 we have also now re-labeled y-axis of the bar graphs as “% of CD45⁺CD11b⁺ Siglec-F⁻ cells” to make the connection to the representative dot plots more clear (please note that the cells were thus pre-gated both for CD45 and CD11b). Moreover, we now provide a new Supplementary Figure 1 showing representative examples of all different gating strategies used in this study. **(lines 64-71 (Results), 405-407 (Methods), lines 664-666, 678, 697, 740, 766, 814, 834, and 865 (Figure legends), new Supplementary Figs. 1a-d)**

Also, gating strategy of FACS analysis is also incorrect. (this is caused by using fewer markers). For example, in fig 1d, authors stated that they showed F4/80 int and F4/80 hi macrophage

population in mammary gland in WT and Plvap^{-/-} mice. However, in the fraction of F4/80^{int} (black line), F4/80^{hi} population was included. Similar ambiguous analysis was performed in Fig. 2d. Since theme of this manuscript is differentiation system of macrophages in mammary gland, a precise FACS analysis should be performed.

We have now re-analyzed all FACS data using the precise gating strategy. We re-checked the positions of the gates in all original FACS plots. In each experiment the F4/80^{hi} and F4/80^{int} gates were set based on the populations seen in wild-type mice, and the same gates were then applied for the knock-out and antibody-treated mice. Notably, the refining of the gating and re-analyses of all data only slightly altered the percentages of different macrophage populations, but did not alter in any way the main results or conclusions. **(all Figures)**

In fact, one of the major points in this manuscript is to demonstrate that F4/80^{hi} and F4/80^{int} expression, although very commonly used in the literature, is not the optimal marker for defining the different macrophage populations. We show that utilization of CD206 expression instead allows much more clear-cut distinction of the two populations (please see Fig. 3a, Fig. 4a, and Fig 5a and b).

Reviewer #2 (CyToF, Systems immunology, immune cell subsets)(Remarks to the Author):

We would like to thank the reviewer for the positive and constructive comments.

In this study, the authors use a large variety of mouse model systems to study the origins of mammary gland resident monocyte/macrophage populations. They show that F4/80^{hi} cells are enriched for fetal derived macrophage cells that can be better defined using CD206 as marker based on mass cytometry data analysis performed. This appears to be one of the most rigorous studies of mammary gland resident monocyte macrophage steady-state ontogeny and should be broadly useful. Overall, my impression is that this is a well-executed study and I do not have any major problems.

Comments:

On line 65, the justification on the origins of these cell populations are not provided at this point. Instead perhaps objective descriptions should be used and origins stated as a hypothesis?

We have modified the text as suggested by the referee. We now describe the observed objective phenotypes of mammary gland macrophages first, and then state the possible different origins of F4/80^{hi} and F4/80^{int} macrophages as a hypothesis. **(lines 68-76, 86-90 and 407-412)**

Paragraph starting on line 83. If possible a reference for the validity of this fate mapping model should be provided.

We now describe the development and validity of the genetic fate-mapping models in more detail and provide references for them. **(lines 94-97 and 99-106)**

In general, figure legends need more details to allow them to be properly interpreted. Fig. 1. All upstream gating should be described in figure legends for each panel. Panel b, the colors and

backgating strategies are not at all described. What is the basis for coloring in Fig. 3e? Are these manual or automatically defined clusters?

We sincerely apologize for describing our gating strategy too superficially in the Methods and Figure legends of the original submission. We have now specified the macrophage gating for each figure panel more precisely in the Figure legends. In addition, we provide a new Supplementary Figure showing representative examples of all different gating strategies used in this study. In this Supplementary Figure, specific reference to each figure panel using the given gating is also provided. **(lines 664-666, 678-681, 697-700, 733-735, 740, 766, 814, 834, and 865 (each Figure legend), new Supplementary Figs. 1a-d)**

We have now described the colors and backgating strategies for Fig. 1b both in the Results, Methods and in the Figure legends. **(lines 100-102, 419-423, 669-673, and 808-811)**

We have also explained more carefully what was done in Fig. 3e. We now specify that the clusters are automatically defined (unsupervised hierarchical clustering). Thereafter, we searched for the key leukocyte differentiation markers expressed in each cluster and based on these analyses then manually added the colors for each cluster. **(lines 219, 222-223, 460-462, 719-722, 882-884 new supplementary Figs. 6a and b)**

Error in panel label on Fig S3b – one panel label missing in the pdf I have.

We have amended the missing panel label **(now Fig. S4a)**.

When mass cytometry is introduced, it would be easier to follow if similar gating strategies on the previously quantified F4/80^{int} and F4/80^{hi} cells were annotated on the tSNE plots in addition to these additional subsets defined in Fig. 3b. To make this part of the paper interpretable in light of the data shown in Fig. 1 and 2, it is important that equivalent gating strategies are used for mass cytometry as for the quantifications of the fluorescent data shown in previous figures. Although this does make sense and subsequent figures support the utility of CD206 as a marker for embryonic-derived macrophages, biaxial dot plots (with mass cytometry F4/80 staining) should then be used to support lines 173-174.

We now provide the new gating data requested by the referee:

The F4/80^{int} and F4/80^{hi} cells on the SNE plots have now been annotated. Please note that one of the main findings of the paper is that F4/80^{int} and F4/80^{hi} are not performing optimally in defining the two macrophage populations (and that CD206 gives more discrete populations). **(lines 860-862, new Supplementary Fig. 5b)**

We now also provide data from manual bi-axial gating of the mass cytometric data using the same gating strategy as used in Figs 1 and 2 (viable CD45⁺SiglecF⁻ cells plotted for CD11b and F4/80). In the same context we provide biaxial dot plots from the mass cytometry F4/80 stainings for CD206. **(lines 449-451, new Supplementary Fig. 5a)**

Collectively, all these analyses together with the fate-mapping experiments support the notion that after birth the embryonic-derived macrophages are F4/80^{hi}CD206⁺ and adult derived macrophages are F4/80^{int}CD206^{neg/low}, and that CD206 expressions allows more clear cut definition of these populations than does F4/80 expression.

Although it is nice that the mass cytometry data could be used as a justification for the use of CD206, I wonder if this could have been deduced in a more objective way from these data. For instance, perhaps the authors could use high dimensional clustering analysis (as in Fig. 3e) to first define the most-differentiated macrophage, the use an automatic gating algorithm to objectively determine the most accurate way to define these cells. This is just a suggestion and not necessary for the validity of this manuscript.

We apologize for not being clear enough in explaining our search strategy for the best marker defining the different macrophage populations, which was actually done exactly as the referee suggests. Thus, the different myeloid cell (CD45⁺CD11b) populations in Figs 3a-c have been generated using unsupervised hierarchical clustering programs. We then used the consensus markers for neutrophils (Ly6G), eosinophils (Siglec-F) and dendritic cells (CD11c) to define these non-macrophage clusters. Thereafter, we used F4/80 expression, which has been reported in numerous publications to accurately define the embryonic-derived and adult-derived macrophage populations, to verify the location of these F4/80^{hi} and F4/80^{int} cells within the macrophage cluster. Finally, we used the remaining monocyte/macrophage markers of our panel (CD206, Ly6C, MHCII, CD68 etc) to search for a potential marker, which would give more clear-cut separation of the embryonic (F4/80^{hi}) and adult (F4/80^{int}) -derived macrophages. These analyses defined CD206 as the most powerful marker (within our 25 marker panel) for separating the two ontogenically different macrophage populations. The experimental rationale has been now explained more carefully. **(lines 183-188, 443-448, 707-712, 841, 842-847)**

For Fig. 3B, biaxial scatter plots should be shown for the gating strategies used for each of these manually defined subsets.

The different clusters in Fig.3b were generated using unsupervised hierarchical clustering, and their identity was based on the expression of the consensus markers shown in the Figure (please see above for details). Therefore, we believe that biaxial scatter plots will not necessarily provide additional information, but as an example, we now show here how manual gating of the different clusters in 5 wk old MG, would look like for the biaxial gating of the markers used in the panel. If requested, we are happy to include this (and a similar plot from the 3 month old MG) as additional Supplementary Figures.

Figure for reviewer

For the recovery of F4/80^{hi} cells after anti-CSF-1+clodronate liposome treatment (figs 2,4), did the authors go beyond 11 days of recovery to see if these cells can eventually recover?

We didn't go beyond 11 days of recovery in these experiments. We fully agree with the referee that it would be interesting to see if this population recovers in 12 wk old (fully adult) mice, but the experiment would not be compatible with the 3 mo time limit for the revisions. However, we have now discussed this aspect in more detail (**lines 307-311**).

REVIEWERS' COMMENTS:

Reviewer #1 (Remarks to the Author):

The authors adequately addressed my comments.

Regarding gating strategy, it needs some points corrected for readers. For example, in fig1a or supplementary fig1a, authors showed CD45+, siglec-f-, Mac1+ population were analyzed by using F4/80 and CD45 to detect macrophages. CD45 was used in the first gating, therefore authors should not use it again. Moreover, "fetal liver derived" population and "yolk sac derived" population was very closed. Thus, it is better to analyze the population (CD45+, siglec-f-, Mac1+) with F4/80 and a marker other than CD45.

Reviewer #2 (Remarks to the Author):

My comments have been adequately addressed. Clarifications and additional representative plots are much appreciated. I have two minor concerns remaining:

Line 443-444, viSNE is not an unsupervised hierarchical clustering program – it is an implementation of tSNE, a dimensionality reduction data visualization method. How was unsupervised hierarchical clustering performed then?

For each figure panel, please specify which data are from $n = 1, 2$ or 3 independent experiments. The labeling now is overly vague. In many plots, there appear to be more points than the n values specified – please clarify.

REVIEWERS' COMMENTS:

Reviewer #1 (Remarks to the Author):

We thank the reviewer for the constructive comments.

The authors adequately addressed my comments.

Regarding gating strategy, it needs some points corrected for readers. For example, in fig1a or supplementary fig1a, authors showed CD45+, siglec-f-, Mac1+ population were analyzed by using F4/80 and CD45 to detect macrophages. CD45 was used in the first gating, therefore authors should not use it again. Moreover, “fetal liver derived” population and “yolk sac derived” population was very closed. Thus, it is better to analyze the population (CD45+, siglec-f-, Mac1+) with F4/80 and a marker other than CD45.

We agree that our gating strategy is relatively complicated. The reasons for this are as follows:

1. We first wanted to restrict our analyses to leukocytes only, and therefore used CD45 right in the beginning of the gating process. Thereafter, we excluded eosinophils based on Siglec-F expression, since eosinophils also express F4/80. We then focused our analysis on myeloid cells (identified by Mac-1/CD11b expression), which include our target population (monocytes and macrophages). Based on the literature, F4/80 was finally chosen as the “prototype” marker for separating fetal liver-derived and yolk sac -derived macrophages. **(lines 67-69 and 412-415, in orange)**
2. For illustrative purposes we found that plotting F4/80 against CD45 gives visually the best separation between F4/80 positive and F4/80 intermediate macrophage populations. Although CD45 is used earlier in the gating strategy, it does not bias the results in any way. All quantitative data are given as percent of CD45+SiglecF-CD11b+ cells, and thus only cells which are included in **all** previous gateings have been analyzed. **(new text on lines 419-422, in orange)**
3. As the reviewer correctly points out, even when choosing F4/80 vs. CD45 for the dot plots, the different F4/80 populations are very close to each other. Actually, we initially screened several other myeloid markers (CD64, MerTEK, Ly6C, MHCII) to see if their combination to F4/80 gave better resolution of F4/80 high and F4/80 intermediate populations. However, they all were inferior to CD45. Therefore, the majority of our animal experiments shown in Figs 1 and 2 was done with an antibody pool, which only contained CD45, Siglec-F, CD11b and F4/80. Based on the superior performance of CD45, we always reserved the same fluorochrome channel for it in all experiments, including the different cell fate mapping experiments. Therefore, analyzing our data with CD45 vs. F4/80 minimizes interexperimental variation. Due to this strategy, we unfortunately do not have data available for choosing a marker other than CD45 for the most experimentation presented in Figs 1 and 2. **(lines 423-426, in orange)**
4. Finally, as foreseen by the reviewer, we fully agree that F4/80 vs. CD45 plotting is not performing optimally. This is exactly the reason, why we went on and performed the unbiased single cell phenotyping mass cytometry experiments, which led to the discovery of CD206 as a better discriminating phenotypic marker. We thoroughly validated the performance of CD206 in comparison to F4/80 in Fig.4 (panels a-f). Therefore, one of the major conclusions of the paper is that for future flow cytometric studies of mammary gland macrophages it is advisable to use a pool containing at least CD45, Siglec-F, CD11b, F4/80 and CD206, which will also allow the use of F4/80 vs. CD206 as the final plot for illustrative purposes. **(lines 292-295, in orange)**

Reviewer #2 (Remarks to the Author):

My comments have been adequately addressed. Clarifications and additional representative plots are much appreciated. I have two minor concerns remaining:

Line 443-444, viSNE is not an unsupervised hierarchical clustering program – it is an implementation of tSNE, a dimensionality reduction data visualization method. How was unsupervised hierarchical clustering performed then?

We thank the reviewer for pointing out this important point. We have now reworded the text so that it is clear that viSNE is a data visualization program, which segregates cells into spatially distinct regions based on the markers expressed. We have now replaced all “cluster” words in this context with “cell population” words (**text related to Figs 3a-c, and Supplementary Figs 4a-d and 5a,b and d**).

The unsupervised hierarchical clustering was performed by using the X-shift algorithm run in the Vortex visual space. It estimates the number of cell clusters in high dimensional data by using weighted k-nearest-neighbor density estimation. (**text related to Fig. 3e and Supplementary Fig. a-c**)

For each figure panel, please specify which data are from n= 1, 2 or 3 independent experiments. The labeling now is overly vague. In many plots, there appear to be more points than the n values specified – please clarify.

The numbers of independent experiments (i.e. experiments performed on different days) is now specified panel-by-panel in the Figure legends, as requested. The data points overlaid in the plots/bar graphs represent individual mice. Since we always used several mice/group in each independent experiment, the number of data points is always much higher than the number of independent experiments. In addition to the dots in the plots, the exact number of individual mice is seen from the Source files.